# Non-uniform Bid-scaling and Equilibria for Different Auctions: An Empirical Study

## ABSTRACT

In recent years, the growing adoption of auto-bidding motivates the study of auction design with value-maximizing auto-bidders. It is known that under mild assumptions, uniform bid-scaling is an optimal bidding strategy in truthful auctions, e.g., Vickrey–Clarke–Groves auction (VCG), and the price of anarchy for VCG is 2. However, for other auction formats like First-Price Auction (FPA) and Generalized Second-Price auction (GSP), uniform bid-scaling may not be an optimal bidding strategy, and bidders have incentives to deviate to adopt strategies with non-uniform bid-scaling. Moreover, FPA can achieve optimal welfare if restricted to uniform bid-scaling, while its price of anarchy becomes 2 when non-uniform bid-scaling strategies are allowed.

All these price of anarchy results have been focused on welfare approximation in the worst case scenarios. To complement theoretical understandings, we empirically study how different auction formats (FPA, GSP, VCG) with different levels of non-uniform bid-scaling perform in an auto-bidding world with a synthetic dataset for auctions. Our empirical findings include:

- For both uniform bid-scaling and non-uniform bid-scaling, FPA is better than GSP and GSP is better than VCG in terms of both welfare and profit;
- A higher level of non-uniform bid-scaling leads to lower welfare performance in both FPA and GSP, while different levels of non-uniform bid-scaling has no effect in VCG.

Our methodology of synthetic data generation may be of independent interests.

## ACM Reference Format:

Anonymous Author(s). 2024. Non-uniform Bid-scaling and Equilibria for Different Auctions: An Empirical Study. In *Proceedings of ACM Conference (Conference'17)*. ACM, New York, NY, USA, 11 pages. https://doi.org/XXXXXXX.XXXXXXX

## 1 INTRODUCTION

The online digital advertising market has seen tremendous growth in recent years, with projection to reach \$271.20 billion dollars in United States in 2023 [33]. Along with the historical growth in revenue, the market has also witnessed significant shifts in the bidding behavior model of the advertisers, from manual bidding to auto-bidding [3, 7]. Auto-bidding allows the advertisers to delegate their bidding tasks to autobidding agents by specifying high-level

bidding objectives and constraints. The autobidding agents bid on behalf of the advertisers to procure online advertising opportunities.

Such a shift fundamentally changes how advertisers participate and bid in auctions, driving the system towards a new equilibrium with potentially very different performance in both revenue and welfare. In the classic auction model of VCG auctions with manual bidding (usually modeled as utility maximizers who maximize their quasi-linear utility given by the difference between value and payment), reporting values truthfully is a dominant strategy. However, using values as bids is no longer an optimal strategy for auto-bidding agents (typically modeled as value maximizers who maximize their total value subject to a return-on-investment (ROI) constraint [7]), and therefore the equilibrium outcome may not be efficient. Such an observation motivates a recent line of research on understanding the price of anarchy of different auction formats under auto-bidding. Aggarwal et al. [3] shows that the price of anarchy of any auctions that are truthful for quasi-linear utility maximizers (such as VCG auctions) is 2. Interestingly, it turns out that the price of anarchy for first-price auctions (FPA) is also 2 under auto-bidding [18, 30] and the price of anarchy for the generalized second-price (GSP) auctions has been investigated recently in [17].

However, these theoretical results make two assumptions that are unlikely to hold in practice. First, price of anarchy measures the welfare performance of the worst case equilibrium from the worst case instance, while the welfare performance of equilibria from real-world instances could be much better than the theoretical number. Second, the bidding agents are assumed to bid optimally in response to other bidders' bidding strategies and their bidding profile forms an equilibrium, despite of the fact that finding optimal bidding and/or equilibrium could be computationally intractable.

Aggarwal et al. [3] demonstrates that uniform bid-scaling (i.e., always bid $\kappa v$ with a universal bid-scaling factor $\kappa$ when the value is $v$) is an optimal strategy for value maximizers in auctions that are truthful for quasi-linear utility maximizers. Therefore, each autobidding agent is only required to optimize one the bid-scaling factor to find the best strategy. On the other hand, for auctions that are not truthful for quasi-linear utility maximizers (such as FPA and GSP), uniform bid-scaling can lead to a suboptimal bidding strategy, while non-uniform bid-scaling (i.e., use different bid-scaling factors in different auctions) may greatly improve the bidding performance.

## 1.1 Our Contributions

In this paper, we make two main contributions to the growing literature on auction and bidding in the autobidding world:

### 1.1.1 An experimental framework for analyzing auction/bidding in an autobidding world.
In Section 3, we propose a framework of generating synthetic ad auction data and simulating auctions and bidding algorithms in an autobidding world.

Our synthetic data generation captures key features of real-world application to retain the heterogeneity of the problem. This is particularly important for empirical studies in the auto-bidding world, as independence and symmetry may lead to almost no efficiency loss in certain auction formats, which, however, are known to be inefficient according to prior theoretical and empirical results [4, 20].

In order to efficiently simulate non-uniform bid-scaling strategies, we propose an efficient approximate algorithm to optimize the non-uniform bid-scaling factors iteratively, whose time complexity is linear in the degree of freedom of these strategies.

In addition to that, we also propose a hierarchical structure over the ad auction instances so that we can compare non-uniform bid-scaling strategies of different degrees of freedom.

*1.1.2 Experimental results complementing existing theoretical work on autobidding.* With the data generation and simulation framework, in Section 4, we analyze different auction formats: FPA, GSP, VCG (with/without reserve prices). For each auction format, we simulate the uniform bid-scaling strategy and three non-uniform bid-scaling strategies with different degrees of freedom and achieve reasonable convergence. The total scale of the experiment is about 4.5 trillions of simulation runs.

Compared with existing theoretical work, our empirical results either (1) confirm the results empirically and show they are robust when user costs are introduced or (2) give more insights and understanding for settings in which theoretical worse-case analyses are not capable for comparisons among different auction formats.

## 1.2 Additional Related Work

Our work is closely related to the recently growing body of literature on auction and bidding design in the autobidding world. For improving price of anarchy via mechanism design, Deng et al. [20] present additive boosts based on advertisers' values, Balseiro et al. [4] leverage reserve prices derived from machine-learned predictions, and Liaw et al. [30], Mehta [32] incorporate randomization into the auction rules. On the bidding side, there is a series of work for online bidding with budget constraints [5, 8–10, 24], with both budget and ROI constraints [6, 25, 26, 31], and their variants in the context of multi-channel bidding [16, 34].

## 2 PRELIMINARY

We let $N$ denote the set of advertisers and $M$ denote the set of ad queries[1] where ad slots will be sold through position auctions [22, 35]. Each query $j \in M$ has $z_j$ slots. In this paper, we use $i$ to index advertisers, $j$ to index queries, and $k$ to index slots; we also use $-i$ to indicate advertisers other than advertiser $i$. Each slot $k$ in query $j$ is associated with a *slot click-through rate* $\beta_{j,k} \in [0, 1]$ such that $\beta_{j,k} \geq \beta_{j,k+1}$ for all $1 \leq k < z_j$. For query $j$, advertiser $i$ has a valuation of $v_{i,j}$ for each click and the value that advertiser $i$ receives when winning slot $k$ in query $j$ is $\beta_{j,k} \cdot v_{i,j}$.

*Auction.* The ad slots of all queries are sold by separately conducting some auction mechanism $\mathcal{A}$. An auction mechanism $\mathcal{A}$ specifies an allocation rule and a payment rule that map a bidding

profile $b_j = (b_{1,j}, \ldots, b_{N,j})$ and a list of slot click-through rates $\beta_j$ (of any length) to an allocation and a payment vector, respectively:

$$\text{allocation} : \mathbb{R}_+^N \times [0, 1]^* \to [0, 1]^N, \quad \text{payment} : \mathbb{R}_+^N \times [0, 1]^* \to \mathbb{R}_+^N,$$

where an allocation to each advertiser is either $\beta_{j,k}$ for whom gets allocated the slot $k \in [z_j]$ or 0 for whom does not win the auction. Note that a valid allocation must allocate each slot to at most 1 advertiser, and each slot $k$ can be allocated only if all higher slots $k' < k$ are allocated. In other words, an allocation rule may allocate a prefix of slots, or even none of the slots.

*User cost.* Showing ads to users may have implications on the user experience [1, 11, 12, 19, 21]. Denote $\text{cost}_{i,j}$ the normalized cost of showing advertiser $i$'s ad in auction $j$, where the realized cost will be scaled by the actual allocation, i.e., $\text{cost}_{i,j} \cdot \text{allocation}_{i,j}$.

*Value Maximizers.* We model each advertiser $i$ as a value maximizer with an ROI constraint [7] who aims to maximize the total value subject to a constraint that the ratio between the total value and the total payment should not be below a given threshold $\tau_i$. Formally, given other bidders' bidding strategy $b_{-i}$ each advertiser $i$ optimizes their bidding strategy to maximize the following program

$$\max_{b_i \in \mathcal{B}_i} \quad \sum_{j \in M} v_{i,j} \cdot \text{allocation}_{i,j}$$
$$\text{subject to} \quad \sum_{j \in M} v_{i,j} \cdot \text{allocation}_{i,j} \geq \tau_i \cdot \sum_{j \in M} \text{payment}_{i,j}, \quad (1)$$

where $\mathcal{B}_i$ is a set of feasible bidding strategies, $\text{allocation}_{i,j} = \text{allocation}_i(b_j, \beta_j)$ is the allocation of advertiser $i$ in query $j$, and similar for $\text{payment}_{i,j}$. Denote the total value and payment as $\text{value}_i = \frac{1}{\tau_i} \sum_{j \in M} v_{i,j} \cdot \text{allocation}_{i,j}$ and $\text{spend}_i = \sum_{j \in M} \text{payment}_{i,j}$, we rewrite the program (1) as:

$$\max_{b_i \in \mathcal{B}_i} \quad \text{value}_i$$
$$\text{subject to} \quad \text{value}_i \geq \text{spend}_i. \quad (2)$$

*Welfare and Profit.* We measure the performance in terms of welfare and profit. With $\text{cost}_i = \sum_{j \in M} \text{cost}_{i,j} \cdot \text{allocation}_{i,j}$, define

$$\text{welfare} = \sum_{i \in N} \text{value}_i - \text{cost}_i,$$
$$\text{profit} = \sum_{i \in N} \text{spend}_i - \text{cost}_i.$$

*Uniform Bid-scaling.* A uniform bid-scaling strategy can be described by a bid multiplier $\kappa_i \in \mathbb{R}_+$ for advertiser $i$: $b_{i,j} = \kappa_i \cdot v_{i,j}/\tau_i$.

*Non-uniform Bid-scaling.* In general, any bidding strategy that is not a uniform bid-scaling strategy is a non-uniform bid-scaling. Hence, advertiser $i$ can choose different bid multipliers for each query $j$, i.e., $\kappa_{i,j} \in \mathbb{R}_+$. However, implementing such general non-uniform bid-scaling strategies rely on the unrealistic complete knowledge of other advertisers on each query, thus intractable for both simulation and practical applications. In this paper, we define partitions to the queries, and for each partition $d$, a non-uniform bid-scaling strategy chooses one bid multiplier $\kappa_{i,d} \in \mathbb{R}_+$.

In addition, we are interested in how the simulation results vary with the granularity of the partition. Hence we introduce a multi-layer partition of the query set: Suppose there is a $L$-layer laminar set family $\mathcal{S}$ defined on top of the query set $M$. There are $q_\ell$ sets for layer $\ell \in [0, 1, \cdots, L]$ and $S_{\ell,d} \subseteq M$ denotes the $d$-th set in layer $\ell$.

*Definition 2.1.* $\mathcal{S}$ is an $L$-layer laminar family if $\mathcal{S}$ satisfies the following properties:

---

[1]An ad query refers to an event when an ad auction system receives a request to determine a set of ads to present to a user. For example, when a user opens a web page with ad display slots, an ad query is created.

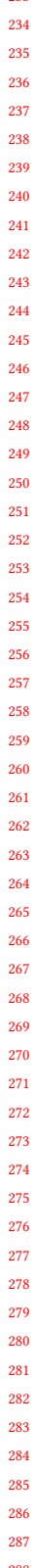

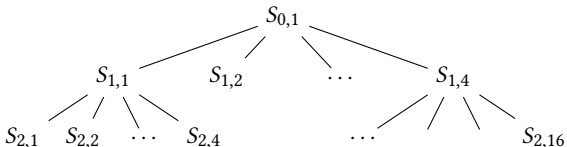

**Figure 1: An example of a 2-layer laminar set family. For each node, its children sets form a partition of it. Sets in the same layer have no intersection.**

- $q_0 = 1$;
- For all layer $\ell$, $\bigcup_{d=1}^{q_\ell} S_{\ell,d} = M$;
- For all layer $\ell$ and all $1 \le a < b \le q_\ell$, $S_{\ell,a} \cap S_{\ell,b} = \emptyset$;
- For all $\ell_1 < \ell_2$, $1 \le a \le q_{\ell_1}$, and $1 \le b \le q_{\ell_2}$, either $S_{\ell_2,b} \subseteq S_{\ell_1,a}$ or $S_{\ell_1,a} \cap S_{\ell_2,b} = \emptyset$.

Intuitively, in an $L$-layer laminar set family $\mathcal{S}$, each layer $\ell$ constitutes a partition of $M$ and a deeper layer (i.e., a layer with a larger layer id $\ell$) gives a finer partition of $M$. See Figure 1 for an example.

We are now ready to define feasible non-uniform bid-scaling strategies based on a given $L$-layer laminar set family $\mathcal{S}$.

*Definition 2.2.* For a $L$-layer laminar set family $\mathcal{S}$, the feasible set of $\ell$-level non-uniform bid-scaling strategies for advertiser $i$ is

$$\mathcal{B}_i^{\mathcal{S},\ell} = \left\{ \boldsymbol{b}_i \in \mathbb{R}_+^M \mid \forall j \in S_{\ell,d}, b_{i,j} = \kappa_{i,d} \cdot v_{i,j}/\tau_i \right\}.$$

Note that the set of feasible 0-level non-uniform bid-scaling strategies corresponds to the uniform bid-scaling strategy with a universal bid-scaling factor $\kappa_i$ for advertiser $i$.

## 3 SIMULATION SETUP

In this section, we describe the setup of the simulation system, including how the auction data is generated, how auctions are implemented, and how the bidding algorithm works.

### 3.1 Data generation procedure

One goal of the data generation design is to mimic the data structure of practical ad auctions, while not involving any real-world empirical knowledge. One key motivation behind this approach is that one can elicit the properties of the problem rooted in its data structure while being robust to the details of the distributions.

We illustrate a model that captures the basic elements in a multi-slot ad auction. In particular, Example 3.1 introduces the typical steps of one auction. We note that some of the terminologies in this example may not simply correspond to the notions we defined in Section 2, and we will explain in details in Section 3.1.1 how the terminologies in this example are mapped to our model.

*Example 3.1 (Life cycle of one auction).* An ad auction starts when a user makes a request to the platform, which can be a concrete query with a specific goal (keyword-based ad auction) or a general ask to fill in ad slots of a web-page (keyword-less ad auction). If the platform considers this request has enough commercial value, then the platform will starts to collect advertisers to participate in an auction to determine which ads to present to the user.

*Candidate retrieval.* Based on the characteristics of the request, the platform needs to retrieve a subset of candidate advertisers that have a decent match with the request. This step can reduce the latency of the later auction stage by limiting the number of candidates and also ensure that only relevant ads are involved.

*Signal preparations.* For the retrieved candidate ads, the platform and the bidding system compute various signals using their ML models, respectively. Example signals include the predicted click-through-rate, predicted conversion rate, etc.

*Bidding.* The auto-bidding system then needs to determine a cost-per-click bid for each ad candidate using this service. For the commonly used target CPA auto-bidding model, the simplest bidding formula will combine the predicted conversion rate (pCVR), the advertiser-set target CPA (tCPA), and the bid multiplier ($\kappa$) for the auto-bidding algorithm to meet the target overtime.

$$\text{bid} = \kappa \cdot \text{pCVR} \cdot \text{tCPA}.$$

*Auction.* Upon receiving the bids from the candidate, the auction needs to select the winner(s) based on the combination of the bids, predicted click-through-rates (pCTR), and user costs (cost). The ad candidates are then ranked by the following score in the decreasing order, and only the first $z$ ad(s) get allocated accordingly.

$$\text{score} = \text{pCTR} \cdot \text{bid} - \text{cost}.$$

Only the ad gets clicked will be charged, where the per-click price may depend on the slot click-through-rate $\beta_k$. The probability of being clicked when shown in slot $k$ is $\beta_k \cdot \text{pCTR}$. The actual price dependency on $\beta_k$ differs by the auction mechanism being used.

*3.1.1 Terminology Mapping.* From the example above, we can summarize the key information needed to run an auction as Table 1, where we also show how each of them corresponds to the notations in our theoretical model. In particular, the bid multiplier $\kappa$, the slot click-through-rate $\beta_k$, and the user cost cost are the same as those in our model, tCPA corresponds to the inverse of the ROI target $\tau$.

The bid in this example corresponds to the per-click bid, while $b$ in our model corresponds to the per-impression bid, i.e., $b = \text{pCTR} \cdot \text{bid}$. This is caused by the separation between the bidding algorithm and the auction, where the bidding algorithm only observes pCVR and the auction only observes pCTR. We note that this separation makes no difference in terms of the mathematical model of the problem. While in practice, since the prediction signal pCTR will not be 100% accurate, the expected payment will depend on the real click-through-rate rather than the prediction pCTR. However, in this work, we focus on the simulation system where the prediction is exactly accurate. Hence we will not treat them separately.

For the same reason, pCTR and pCVR together correspond to the advertiser value $v$, i.e., $v = \text{pCTR} \cdot \text{pCVR}$. Finally, the ad candidate set from retrieval and the score for ranking the candidates are intermediate terms that only exist inside the auction, and hence no corresponding notation in our theoretical model.

Thus, to construct an auction instance, we will need to generate:

- The ad candidate set;
- $v = \text{pCVR} \cdot \text{pCTR}$ for each ad candidate;
- cost for each ad candidate;
- $\beta_k$ for each ad slot (except that $\beta_1 = 1$).

Across all the auction instances, we still need to generate:

- $1/\tau = \text{tCPA}$ for each advertiser.

| Information | Source | Notation |
|---|---|---|
| Ad candidate set | Candidate retrieval | N/A |
| $\kappa$ | bidding system control loop | $\kappa$ |
| pCVR | Bidding system prediction | $v/\text{pCTR}$ |
| tCPA | Advertiser set value | $1/\tau$ |
| bid | $\kappa \cdot \text{pCVR} \cdot \text{tCPA}$ | $b/\text{pCTR}$ |
| pCTR | Auction system prediction | $v/\text{pCVR}$ |
| cost | Auction system prediction | cost |
| $\beta_k$ | Auction system prediction | $\beta_k$ |
| score | score = pCTR $\cdot$ bid $-$ cost | N/A |

**Table 1: Key information in an auction and notations.**

The bid-scaling multiplier $\kappa$ will be computed by the bidding algorithm, which we will discuss in details in Section 3.3.

*3.1.2 Feature Vector Generation.* One key challenge of generating the synthetic data for auction simulation is to avoid symmetry among bidders. Otherwise, if the system converge to a symmetric equilibrium where everyone has roughly the same bid-scaling factors, then all auctions will result in efficient allocations, which, however, can be fundamentally different from practice as real world does not guarantee symmetry.

In practice, the asymmetry comes from the heterogeneity of the features of queries and bidders, and the relationship between the query feature and bidder feature jointly result in different signal predictions, such as pCTR, pCVR, etc.

In our framework, we randomly draw query features and bidder features from multidimensional Gaussian distributions:

$$f_{\text{query}} \sim \mathcal{N}(\mu_{\text{query}}, \Sigma_{\text{query}}), \quad f_{\text{bidder}} \sim \mathcal{N}(\mu_{\text{bidder}}, \Sigma_{\text{bidder}}).$$

Given $f_{\text{query}}$ and $f_{\text{bidder}}$, we generate pCTR and pCVR as:

$$\text{pCTR} \cdot \text{pCVR} = v = \exp(\langle f_{\text{query}}, f_{\text{bidder}}\rangle + \epsilon),$$

where $\langle \cdot, \cdot \rangle$ stands for the inner product of two vectors and $\epsilon$ is a per query Gaussian noise. Hence pCTR $\cdot$ pCVR follows a log-normal distribution, which is consistent with the widely adopted assumption in the literature [29].

Here, we note that directly generating pCTR $\cdot$ pCVR makes no difference with separately generating pCTR and pCVR in terms of simulation, because no matter in the formulation of score (which determines allocation) or in the final expected payment calculation, pCTR and pCVR always show up together in the form of their product. This differs from practice in the expected payment, which depends on the real click-through-rate rather than pCTR.

*Hierarchical Structure of Queries.* To facilitate the simulation of non-uniform bid-scaling algorithms, we need to model the structure of queries, which in practice may follow different topics and categories and form implicit clusters. Queries within the same cluster share more similarities hence attracting a similar set of bidders.

We implement the hierarchical structure of queries by following the structure of a given $L$-layer laminar family (e.g., Figure 1). The query feature generation is done for a given $m$-dimensional Gaussian distribution $\mathcal{N}(\mu, \Sigma)$ by repeating the following steps from $\ell = 1$ to $L$ for all $d$, where $(\mu^{0,1}, \Sigma^{0,1}) = (\mu, \Sigma)$:

(1) For each set $S_{\ell,d} \subseteq S_{\ell-1,d'}$ in layer $\ell$, draw a sample $f \sim \mathcal{N}(\mu^{\ell-1,d'}, \Sigma^{\ell-1,d'})$.

(2) Let $f^{\ell,d} = f^{\ell-1,d'} \oplus (f_1, \ldots, f_{m_\ell})$, where $\oplus$ stands for the vector concatenation operator and $f^{0,1}$ is an empty vector.

(3) Compute the marginal distribution of $\mathcal{N}(\mu, \Sigma)$ conditioned on $f^{\ell,d}$, i.e., the distribution of remaining $(m - \dim(f^{\ell,d}))$ dimensions, as $\mathcal{N}(\mu^{\ell,d}, \Sigma^{\ell,d})$, where $\dim(f^{\ell,d}) = m_1 + \cdots + m_\ell$ is the dimension of $f^{\ell,d}$.

Specifically, let $\mu = \mu_1 \oplus \mu_2$ with $\mu_1$ being the first $\dim(f^{\ell,d})$-dimension and $\mu_2$ being the rest. Decompose $\Sigma$ as follows

$$\Sigma = \begin{bmatrix} \Sigma_{11} & \Sigma_{12} \\ \Sigma_{21} & \Sigma_{22} \end{bmatrix},$$

where $\Sigma_{11}$ is the top-left sub-matrix of size $f^{\ell,d} \times f^{\ell,d}$, and $\Sigma_{12}, \Sigma_{21}, \Sigma_{22}$ being the corresponding sub-matrices. Then $\mu^{\ell,d}$ and $\Sigma^{\ell,d}$ are computed as:

$$\mu^{\ell,d} = \mu_2 + \Sigma_{21}\Sigma_{11}^{-1}(f^{\ell,d} - \mu_1), \ \Sigma^{\ell,d} = \Sigma_{22} + \Sigma_{21}\Sigma_{11}^{-1}\Sigma_{12}.$$

With the hierarchical structure defined above, for each query, we first select a set $S_{L,d}$ in layer $L$ at random, and draw $f_{\text{query}}$ as:

$$f_{\text{query}} = f^{L,d} \oplus f, \ \text{where } f \sim \mathcal{N}(\mu^{L,d}, \Sigma^{L,d}).$$

Note that for queries sharing the same ancestor set $S_{\ell,d}$, the first $(m_1 + \cdots + m_\ell)$ dimensions of their feature vectors are the same, i.e., $f^{\ell,d}$. Therefore, the query features we generated naturally form a hierarchy of clusters.

*3.1.3 Candidate Retrieval.* Given the query and candidate features, we simulate the candidate retrieval process by selecting the $N_{\text{retrieval}}$ candidates with the top correlation score $\langle f_{\text{query}}, f_{\text{bidder}}\rangle$ for each query. We also set a minimum threshold for the candidates on the correlation score, so the total number of retrieved candidates can be less than $N_{\text{retrieval}}$ for some of the queries.

*3.1.4 Remaining Elements.* Besides the above, we also randomly generate other exogenous elements mentioned in Table 1, namely, tCPA, cost, and $\beta_k$. For tCPA, we draw once for each candidate from a Pareto distribution, which is inspired by the observation that things like individual income, city population, firm size, etc often follow a power law distribution with $\alpha \in (2, 3)$ [2, 14, 23]. We note that the Pareto distribution in this case is a simplified version by setting the slowly varying function to constant in the continuous power law distribution model.

$$p_{\text{tCPA}}(x) = \frac{\alpha-1}{x_{\min}} \left(\frac{x}{x_{\min}}\right)^{-\alpha},$$

where $\alpha$ and $x_{\min}$ are the parameters of the distribution.

For cost, we independently draw for each pair of query and candidate from a log-normal distribution: cost $\sim \text{Lognormal}(\mu_{\text{cost}}, \sigma_{\text{cost}}^2)$.

For $\beta_k$, we independently draw their decay factors $\beta_{k+1}/\beta_k$ from a uniform distribution, i.e., $\beta_{k+1}/\beta_k \sim \text{Uniform}[\text{low}, \text{high}]$.

*3.1.5 Distribution Parameter Generation.* To avoid having conclusions sensitive to distribution parameters, we randomly draw the parameters of all the distributions mentioned above for each run and average out the results over multiple runs.

Here, we highlight the generation of the covariance matrix of multidimensional Gaussian distributions for generating features.

- Randomly generate a diagonal matrix $D$, where the elements on its diagonal are i.i.d. from a uniform distribution.
- Randomly generate a noise matrix $N$ (the same size of $D$), where each element is drawn independently from a normal distribution $\mathcal{N}(\mu_{\text{noise}}, \sigma^2_{\text{noise}})$. Then scale $N$ such that $\|N^\mathsf{T}N\|$ equals the desired noise level and set $\Sigma = D + N^\mathsf{T}N$.

## 3.2 Auction Formats

We consider three commonly studied auction formats:

- FPA: First-Price Auction [15, 28];
- GSP: Generalized Second-Price auction [22, 35];
- VCG: Vickrey–Clarke–Groves auction [13, 27, 36].

In particular, we are also interested in their variants by introducing *reserves*, as Balseiro et al. [4] suggests that adding value-correlated reserve prices to these auctions can improve the worst case approximation bounds in the auto-bidding world.

We then describe the implementation details of all three auction formats with reserve. Note that the vanilla version without reserve is simply a special case with all reserves being 0.

*3.2.1 Allocation Rules.* In the setting of position auctions, the allocation rules for FPA, GSP, and VCG (with reserve) are exactly the same. Intuitively, three steps (formally, see Algorithm 1):

(1) All advertisers with bids lower than the corresponding reserves will be first excluded;
(2) Then the remaining advertisers are ranked by their score in the descending order;
(3) The first $z_j$ advertisers (if the number of remaining advertisers is smaller than $z_j$, then they are all winners) win the auction and get allocated the slots respectively, according to the order by their score.

---

**Algorithm 1:** The common allocation rule for FPA, GSP, and VCG with reserves

**Data:** Bids $\boldsymbol{b} = \{b_i\}$, reserves $\{\text{reserve}_i\}$, scores $\{\text{score}_i\}$, slot click-through-rates $\beta = \{\beta_k\}$, number of slots $z$.

**Result:** Allocated slot $k_i$ for advertiser $i$, and $\{\text{allocation}_i\}$.

1 **for** $i$ *in* $\{1, \ldots, N\}$ **do**
2    **if** $b_i \geq \text{reserve}_i$ *and* $\text{score}_i \geq 0$ **then**
3       Winners.append($i$);
4    **end**
5 **end**
6 Sort Winners in the descending order of their score;
7 **if** Winners.length() $> z$ **then**
8    Winners $\leftarrow$ Winners.topK($z$);
9 **end**
10 $k \leftarrow 1$;
11 **for** $i$ *in* Winners **do**
12    $k_i \leftarrow k$; $\text{allocation}_i \leftarrow \beta_k$; $k \leftarrow k + 1$;
13 **end**
14 **for** $i$ *in* $\{1, \ldots, N\} \setminus$ Winners **do**
15    $k_i \leftarrow \emptyset$; $\text{allocation}_i \leftarrow 0$;
16 **end**
17 **return** $\{k_i\}$, $\{\text{allocation}_i\}$;

---

*3.2.2 Payment Rules.* The payment rules are different for FPA, GSP, and VCG (with reserve). We describe them one by one.

*FPA payment.* allocation multiplies bid, i.e.,

$$\text{payment}_i = b_i \cdot \text{allocation}_i;$$

*GSP payment.* allocation multiplies the minimum bid to beat the max of reserve and the score of the advertiser $i'$ in the next slot (i.e., $k_{i'} = k_i + 1$), or 0 if no such an advertiser, i.e.,

$$\text{payment}_i = \text{allocation}_i \cdot \max\{\text{reserve}_i, \text{score}_{i'} + \text{cost}_i\};$$

*VCG payment.* Depends on allocation, reserve, cost, and the scores and allocations for all advertisers in lower slots (including the ones who passed their reserves but did not get any slot). Formally, see Algorithm 2. We note that the VCG payment implementation will be wrong if the max with $\text{reserve}_i$ in line 4 and 7 are removed while only taking a max with $\text{reserve}_i$ right before returning the payment.

---

**Algorithm 2:** VCG-with-reserve payment for advertiser $i$

**Data:** $\text{allocation}_i$, $\text{reserve}_i$, $\text{cost}_i$, the ordered list of advertisers in lower slots RunnerUpList (including advertisers passed reserves but not won), and their scores $\{\text{score}_y\}$ and allocations $\{\text{allocation}_y\}$

**Result:** $\text{payment}_i$.

1 alloc $\leftarrow \text{allocation}_i$;
2 $\text{payment}_i \leftarrow 0$;
3 **for** $y$ *in* RunnerUpList **do**
4    $\text{payment}_i \leftarrow \text{payment}_i + (\text{alloc} - \text{allocation}_y) \cdot$ $\max\{\text{reserve}_i, \text{score}_y + \text{cost}_i\}$;
5    alloc $\leftarrow \text{allocation}_y$;
6 **end**
7 $\text{payment}_i \leftarrow \text{payment}_i + \text{alloc} \cdot \max\{\text{reserve}_i, \text{cost}_i\}$;
8 **return** $\text{payment}_i$;

---

## 3.3 Bidding

In our experiments, we implement uniform and non-uniform bid-scaling algorithms. To reach an equilibrium, we simulate 25 rounds of updates for the bidding algorithms to converge.

*3.3.1 Uniform bid-scaling.* For the uniform bid-scaling, we adopt a simple yet effective update strategy by Deng et al. [20]:

$$\log \kappa_{t+1} = (1 - \eta_t) \cdot \log \kappa_t + \eta_t \cdot \log \frac{\text{value}_t}{\text{spend}_t},$$

where $t$ is the index of iteration. The formula above is effectively a gradient descent update to $\log \kappa$ with learning rate $\eta_t$. The stationary point is reached if and only if $\text{value}_t = \text{spend}_t$.

*3.3.2 Non-uniform bid-scaling.* The non-uniform bid-scaling algorithm has to be more sophisticated as one has to explore different combinations of bid-scaling factors on the given partitions of queries. A naive algorithm needs to search exponentially many (in partition size) combinations to find the best one. Here we formulate the problem for a fixed bidder as the following program and propose an efficient approximate solution to it. Note that we omitted

the subscript $i$ to simplify the notation.

$$\max \quad \sum_d \sum_{j \in S_d} \text{allocation}_j(\kappa_d) \cdot v_j$$
$$\text{s.t.} \quad \sum_d \sum_{j \in S_d} \text{allocation}_j(\kappa_d) \cdot v_j \geq \tau \cdot \sum_d \sum_{j \in S_d} \text{payment}_j(\kappa_d)$$

In particular, $\text{allocation}_j(\cdot)$ is the bidder's allocation of auction $j$ as a function of the bid-scaling factors $\kappa_d$ for each partition $d$, where its dependency on the signals, auction format, and competitors are hidden for notation simplicity. Similar for $\text{payment}_i(\cdot)$. With $\{\kappa_d\}$ being the variables, we apply the substitutions

$$\text{value}_d(\kappa_d) = \frac{1}{\tau} \sum_{j \in S_d} \text{allocation}_j(\kappa_d) \cdot v_j$$
$$\text{spend}_d(\kappa_d) = \sum_{j \in S_d} \text{payment}_j(\kappa_d)$$

to rewrite the program as

$$\max \quad \sum_d \text{value}_d(\kappa_d)$$
$$\text{subject to} \quad \sum_d \text{value}_d(\kappa_d) \geq \sum_d \text{spend}_d(\kappa_d).$$

This is in fact a Knapsack problem, where $\text{value}_d(\kappa_d)$ is the "quantity" of the $d$-th good and $\text{spend}_d(\kappa_d) - \text{value}_d(\kappa_d)$ is the corresponding "weight" (non-linear in "quantity"). The goal is to maximize the total "quantity" of selected goods with "capacity" being 0.

Our approximate algorithm (Algorithm 3) has three steps:

(1) Discretize the two functions $\text{value}_d(\kappa_d)$ and $\text{spend}_d(\kappa_d)$ over a finite set of $\kappa_d \in K_d$;
(2) For each partition $d$, compute the lower convex hull of the set of points $\{(\text{value}_d(\kappa_d), \text{spend}_d(\kappa_d)) | \kappa_d \in K_d\}$;
(3) Sort the vertices on the lower convex hulls of each $d$ by their sub-gradient in ascending order and increase the corresponding $\kappa_d$ unless the ROI constraint will get violated.

## 3.4 Parameters and scales of experiments

All of our experimental results come from 100 independent runs. For each run, we generate the distributions as described in Section 3.1.5 and then generated $N = 100$ advertisers and $M = 1000000$ queries. Each of these queries retrieves up to 15 most relevant advertisers to participate in the auction, where each auction has $k = 4$ slots. Within each run, we simulate 25 rounds of updates for the bidding algorithms to converge. Therefore the total number of auctions simulated for each auction format using a uniform bid-scaling strategy is at the order of 2.5 billions. The number of auction simulations with a non-uniform bid-scaling strategy will be much large, because to build the discretization $\{P_d^y\} = \{(\text{value}_d(\kappa_d^y), \text{spend}_d(\kappa_d^y))\}$ as input for Algorithm 3, we will need to simulate once for every pair of advertiser and bid-scaling factor discretization. So the order of auction simulations for each auction format with non-uniform bid-scaling strategy blows up to roughly 250 billions. In the experiment, we simulate 3 levels of non-uniform bid-scaling. Summing up across the auction formats we considered, the total number of auction simulation done is at the order of 4.5 trillions.

## 4 EXPERIMENTAL RESULTS

In this section, we present our experimental results on different auction formats and different non-uniform bid-scaling levels. Additional results about different auction formats with reserves can be found in Appendix A.

---

**Algorithm 3:** Non-uniform Bid-scaling Algorithm

**Data:** Points consist of $\text{value}_d$ and $\text{spend}_d$ on the discretized bid-scaling factors set $\{\kappa_d^y\}$:
$$\{P_d^y\} = \{(\text{value}_d(\kappa_d^y), \text{spend}_d(\kappa_d^y))\}$$
**Result:** One selected point $P_d^*$ for each $d$ corresponding to the selected bid-scaling factors

```
/* Pre-process the points for each partition,
   only keep the ones on the corresponding lower
   convex hull, and sort in ascending order.   */
```
1 **for** each partition $d$ **do**
2 $\quad \{P_d^y\} \leftarrow \text{LowerConvexHull}(\{P_d^1, \ldots, P_d^{K_d}\})$;
3 $\quad$ Sort $\{P_d^y\}$ in the ascending order of its first coordinate;
4 $\quad F_d \leftarrow P_d^1$ /* Initialize the frontier. */
5 **end**
```
/* Greedily push forward the frontier {F_d} on
   each slice along their convex hulls as long
   as the ROI constraint is not violated.     */
```
6 $Q \leftarrow \text{PriorityQueueByAscendingRightGradient}(\{F_d\})$
```
/* The right gradient of F_d is the slope
   between F_d and the next vertex to its right
   on the convex hull.                         */
```
7 **while** $F \leftarrow \sum_d F_d$ is below the $45°$ line **do**
```
   /* The ROI constraint is not violated yet,
      update the best-so-far solution.         */
```
8 $\quad \{P_d^*\} \leftarrow \{F_d\}$;
9 $\quad y \leftarrow \text{PartitionIndexOf}(Q.\text{top}())$;
10 $\quad Q.\text{pop}(F_y)$;
11 $\quad$ **if** $F_y$ is not the last vertex on its convex hull **then**
12 $\quad\quad F_y \leftarrow \text{NextVertexOnConvexHull}(F_y)$;
13 $\quad\quad Q.\text{push}(F_y)$;
14 $\quad$ **end**
15 **end**
16 **return** $\{P_d^*\}$;

---

## 4.1 Uniform bid-scaling in different auction formats

We start by discussing our empirical results of uniform bid-scaling for different auction formats VCG, GSP, and FPA.

Theoretically, under uniform bid-scaling, Aggarwal et al. [3] shows that VCG has PoA of 2 for welfare without cost and Deng et al. [20] shows that FPA gives the optimal welfare and profit. As shown in Table 2 which uses the widely adopted GSP auctions as the benchmark, we observe that FPA > GSP > VCG for both welfare and profit. Our empirical results are consistent with the theoretical finding in the sense that FPA has better welfare and profit. For the comparison between VCG and GSP, theoretical PoA results would not be able to predict the comparison between them in an average case. Our empirical result fills this blank and shows that GSP has better welfare and profit than VCG.

The main argument in Deng et al. [20] to show that FPA gives optimal welfare and profit under uniform bid-scaling is that advertisers bid exactly their value multiplied by bid multiplier 1 in

| Mechanism | Profit | Welfare | Bid Mul |
|-----------|--------|---------|---------|
| GSP | +0.00% (benchmark) | +0.00% (benchmark) | 2.63 [2.31, 2.95] |
| FPA | +4.67% [+3.78%, +5.56%] | +4.60% [+3.73%, +5.48%] | 1.00 [1.00, 1.00] |
| VCG | -2.65% [-3.10%, -2.20%] | -2.59% [-3.04%, -2.15%] | 3.66 [3.14, 4.18] |

**Table 2: Metric comparison among uniform bid-scalings.**

such setting. However, in other auction formats like VCG and GSP, bid multipliers could be larger than 1. When two advertisers with different bid multipliers compete in the same auction, even without cost, their score rankings (i.e. $\kappa_1 \cdot v_1/\tau_1$ vs $\kappa_2 \cdot v_2/\tau_2$) can be different from their value ranking (i.e. $v_1/\tau_1$ vs $v_2/\tau_2$) and this could result in welfare efficiency loss. Intuitively, larger non-uniformity of bid multipliers across advertisers will make the welfare worse.

We check out this intuition empirically and use the average bid multipliers as a proxy for monitoring the non-uniformity of bid multipliers. As shown in the last column of Table 2, we observe that the average bid multiplier in FPA is 1 and the average bid multiplier in GSP is lower than the average bid multiplier in VCG. This is consistent with the intuition.

## 4.2 Non-uniform bid-scaling in different auction formats

In this subsection, we compare different auction formats under (highest-level) non-uniform bid-scaling. As shown in Table 3 which uses GSP as the benchmark, we again observe FPA > GSP > VCG for both welfare and profit.

| Mechanism | Profit | Welfare | Bid Mul |
|-----------|--------|---------|---------|
| GSP non-uniform | +0.00% (benchmark) | +0.00% (benchmark) | 2.62 [2.31, 2.93] |
| FPA non-uniform | +2.81% [+1.92%, +3.69%] | +3.90% [+3.02%, +4.77%] | 1.00 [1.00, 1.00] |
| VCG non-uniform | -3.14% [-3.60%, -2.67%] | -2.26% [-2.66%, -1.85%] | 3.66 [3.14, 4.18] |

**Table 3: Metric comparison among non-uniform bid-scalings.**

In contrast, for PoA of welfare, Aggarwal et al. [3] shows that VCG has PoA of 2 and Liaw et al. [30] shows that no auction formats can get PoA better than 2. These two theoretical results jointly demonstrate that VCG has the optimal PoA. Our empirical results give concrete examples to show that the worst-case optimality of VCG does not be simply generalized to average cases.

## 4.3 Uniform vs non-uniform bid-scaling

Although changing from uniform bid-scaling to non-uniform bid-scaling does not change the ranking of welfare and profit between different auction formats, this change does have different impacts on different auction formats. In this subsection, we compare different non-uniform bid-scaling levels within each auction format.

For FPA, from Table 4, we see clearly that switching to non-uniform bid-scaling hurts welfare and profit. Again, this is consistent with the theoretical result in Deng et al. [20] that FPA combined with uniform bid-scaling gives the optimal welfare and profit.

| Mechanism | Profit | Welfare | Bid Mul |
|-----------|--------|---------|---------|
| FPA uniform | +0.00% (benchmark) | +0.00% (benchmark) | 1.00 [1.00, 1.00] |
| FPA level 1 | -0.25% [-0.28%, -0.22%] | -0.22% [-0.25%, -0.20%] | 1.00 [1.00, 1.00] |
| FPA level 2 | -0.75% [-0.80%, -0.70%] | -0.55% [-0.59%, -0.50%] | 1.00 [1.00, 1.00] |
| FPA level 3 | -1.28% [-1.35%, -1.21%] | -1.02% [-1.08%, -0.96%] | 1.00 [1.00, 1.00] |

**Table 4: Metric comparison among FPA under different non-uniform bid-scalings.**

On the other hand, for VCG, as shown in Table 5, switching to different levels of non-uniform bid-scaling has no effects on welfare and profit. This is consistent with the theoretical results in [3] showing the optimality of uniform bid-scaling in truthful auction.

| Mechanism | Profit | Welfare | Bid Mul |
|-----------|--------|---------|---------|
| VCG uniform | +0.00% (benchmark) | +0.00% (benchmark) | 3.66 [3.14, 4.18] |
| VCG level 1 | -0.00% [-0.00%, +0.00%] | +0.00% [-0.00%, +0.00%] | 3.66 [3.14, 4.18] |
| VCG level 2 | -0.00% [-0.00%, +0.00%] | -0.00% [-0.00%, +0.00%] | 3.66 [3.14, 4.18] |
| VCG level 3 | -0.00% [-0.00%, +0.00%] | -0.00% [-0.00%, +0.00%] | 3.66 [3.14, 4.18] |

**Table 5: Metric comparison among VCG under different non-uniform bid-scalings.**

For GSP, the story is more complicated. From Table 6, we observe that increasing non-uniform bid-scaling level in GSP increases profit but decreases welfare. Welfare and profit going in different directions (or changing by different amount which happened in other tables) is a sign of not converging to a point in which all advertisers meet their target constraints. This could happen in either the benchmark (GSP with uniform bid-scaling) or GSP with non-uniform bid-scaling. We will discuss more in Section 4.4 about convergence. Overall, increasing non-uniform bid-scaling level in GSP does not dramatically change the allocation to make profit and welfare both increase or decrease, but it does make the convergence going more biased towards increasing profit.

Interestingly, in these three tables, if we look at the last columns, the average bid multipliers do not change much for different levels of non-uniform bid-scaling. In order to measure the divergence of the bid multipliers across partitions, we introduce a metric called the *strength* of non-uniform bid-scaling for each bidder:

$$\text{strength} = \text{avg}(|\log \kappa_d - \log \bar{\kappa}|),$$

where $\bar{\kappa}$ is the average bid-scaling factor of that advertiser. The overall strength is the weighted average of the per-bidder strength.

| Mechanism | Profit | Welfare | Bid Mul |
|---|---|---|---|
| GSP uniform | +0.00% (benchmark) | +0.00% (benchmark) | 2.63 [2.31, 2.95] |
| GSP level 1 | -0.02% [-0.04%, -0.01%] | -0.05% [-0.06%, -0.03%] | 2.63 [2.31, 2.95] |
| GSP level 2 | +0.02% [-0.02%, +0.06%] | -0.18% [-0.21%, -0.15%] | 2.62 [2.31, 2.93] |
| GSP level 3 | +0.51% [+0.43%, +0.59%] | -0.35% [-0.40%, -0.29%] | 2.62 [2.31, 2.93] |

**Table 6: Metric comparison among GSP under different non-uniform bid-scalings.**

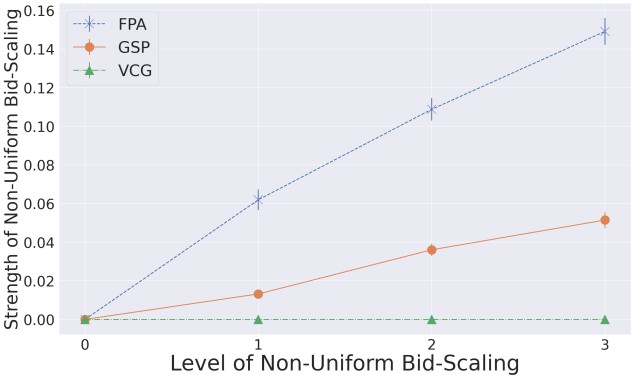

**Figure 2: The strength of non-uniform bid-scaling by the level of non-uniform bid-scaling.**

Note that when a bidder uses uniform bid-scaling, its strength is 0. For each auction format, a higher overall strength would imply that more bidders are actively adopting non-uniform bid-scaling. Figure 2 shows how the strengths of different auction formats evolve as the corresponding level of non-uniform bid-scaling increases. As expected, we see that the strength of FPA increases faster than GSP, while the strength of VCG is always 0 (i.e., uniform bid-scaling).

The trend matches our intuition. The newly introduced strength metric uncovers the divergence of the bid multipliers even though the mean values seem unchanged.

### 4.4 Convergence

In this section, we show empirically how well our auto-bidding algorithms converge to meet the target ROI constraints. This is an important sanity check, since without good convergence, the empirical results could be very badly biased and would not represent what would happen in an equilibrium.

To capture the slackness of the target ROI constraints of all bidders in aggregation, we introduce a metric RelativeMargin, which is essentially the sum of the gap between value and spend for each bidder normalized by the total revenue:

$$\text{RelativeMargin} = \frac{\sum_i |\text{value}_i - \text{spend}_i|}{\sum_i \text{spend}_i}.$$

In particular, RelativeMargin = 0 implies perfectly achieving ROI targets for all bidders.

In Figure 3 and Figure 4, we plot RelativeMargin and average ROI of different iterations of bidding algorithms. Notice that after

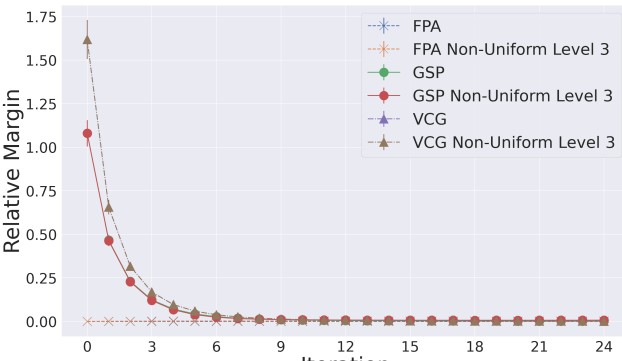

**Figure 3: Relative margin converges to 0 over time.**

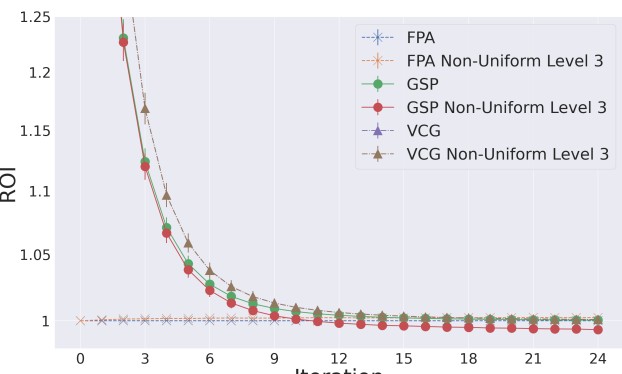

**Figure 4: ROI converges to 1 over time.**

5 to 10 iterations, RelativeMargin drops to a pretty low level and ROI is almost 1, indicating reasonable convergence.

## 5 CONCLUSIONS

To summarize, in this paper, we conduct an empirical study on different auction formats under uniform bid-scaling and non-uniform bid-scaling in an auto-bidding world.

Our main empirical findings are (1) FPA > GSP > VCG in terms of welfare and profit under both uniform bid-scaling and non-uniform bid-scaling, and (2) higher levels of non-uniform bid-scaling has negative impact in FPA but no effect in VCG. Our empirical results complement the theoretical findings from prior work which mainly focus on worst-case type analyses, price of anarchy. We also note that our framework for synthetic auction data generation and running auctions and bidding algorithms in the auto-bidding setup could be of independent interests for future empirical research.

For future work, one direction is to extend our empirical framework to the multi-channel setting studied in [16], in which advertisers procure ad impressions simultaneously on multiple channels and each channel may adopt its own autobidding algorithm, which may implement uniform and/or non-uniform bid-scaling strategies. The multi-channel setting is well-motivated from several practical scenarios but it adds another layer of complexity as the advertisers could potentially set different targets across channels. It would be interesting to see how our empirical observations carry over to the multi-channel setting.

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

## A EXPERIMENTAL RESULTS FOR DIFFERENT AUCTION FORMATS WITH RESERVES

Prior work [4] shows that theoretically VCG with reserves using signals related to values has welfare approximation improves all the way to 1 when signals used in reserves approximate values better and better. [4] also has an empirical section comparing (normal) VCG to VCG with such reserves, under uniform bid-scaling and with no cost. Deng et al. [18] extend [4]'s theoretical results to show a similar result for FPA under non-uniform bid-scaling.

We extend their empirical study to different auction formats, non-uniform bid-scaling, and with costs. In the following three tables (Table 8, 9, 10), we compare reserves versus no reserves, and uniform bid-scaling versus non-uniform bid-scaling in FPA, VCG and GSP separately. The non-uniform bid-scaling analyzed in this section uses the highest non-uniform level in Section 4.2.

For FPA, we observe that our current level of reserves has almost no impact, under both uniform bid-scaling and non-uniform bid-scaling.

For VCG and GSP, we observe that the reserve prices increase both profit and welfare in both uniform and non-uniform bid-scaling. Compared to uniform bid-scaling algorithms, the reserve prices provide higher boost to profit and lower boost to welfare in non-uniform bid-scaling.

For the strength of non-uniform bid scaling and the convergence, see Figure 5.

| Mechanism | Profit | Welfare | Bid Mul |
|---|---|---|---|
| GSP reserve non-uniform | +0.00% (benchmark) | +0.00% (benchmark) | 2.49 [2.23, 2.74] |
| FPA reserve non-uniform | +1.77% [+1.03%, +2.51%] | +3.42% [+2.67%, +4.16%] | 1.00 [1.00, 1.00] |
| VCG reserve non-uniform | -2.46% [-2.85%, -2.07%] | -2.06% [-2.43%, -1.69%] | 3.76 [2.86, 4.65] |

**Table 7: Metric comparison among non-uniform bid-scalings.**

| Mechanism | Profit | Welfare | Bid Mul |
|---|---|---|---|
| FPA uniform | +0.00% (benchmark) | +0.00% (benchmark) | 1.00 [1.00, 1.00] |
| FPA non-uniform | -1.28% [-1.35%, -1.21%] | -1.02% [-1.08%, -0.96%] | 1.00 [1.00, 1.00] |
| FPA reserve uniform | +0.00% [-0.00%, +0.00%] | +0.00% [-0.00%, +0.00%] | 1.00 [1.00, 1.00] |
| FPA reserve non-uniform | -1.32% [-1.39%, -1.24%] | -1.06% [-1.12%, -1.00%] | 1.00 [1.00, 1.00] |

**Table 8: Metric comparison among FPA mechanisms under different reserve and non-uniform bid-scaling settings.**

| Mechanism | Profit | Welfare | Bid Mul |
|---|---|---|---|
| VCG uniform | +0.00% (benchmark) | +0.00% (benchmark) | 3.66 [3.14, 4.18] |
| VCG non-uniform | -0.00% [-0.00%, +0.00%] | -0.00% [-0.00%, +0.00%] | 3.66 [3.14, 4.18] |
| VCG reserve uniform | +0.83% [+0.65%, +1.01%] | +0.84% [+0.66%, +1.02%] | 3.26 [2.92, 3.61] |
| VCG reserve non-uniform | +1.68% [+1.44%, +1.92%] | +0.61% [+0.46%, +0.77%] | 3.76 [2.86, 4.65] |

**Table 9: Metric comparison among VCG mechanisms under different reserve and non-uniform bid-scaling settings.**

| Mechanism | Profit | Welfare | Bid Mul |
|---|---|---|---|
| GSP uniform | +0.00% (benchmark) | +0.00% (benchmark) | 2.63 [2.31, 2.95] |
| GSP non-uniform | +0.51% [+0.43%, +0.59%] | -0.35% [-0.40%, -0.29%] | 2.62 [2.31, 2.93] |
| GSP reserve uniform | +0.51% [+0.38%, +0.64%] | +0.51% [+0.37%, +0.64%] | 2.43 [2.21, 2.65] |
| GSP reserve non-uniform | +1.48% [+1.29%, +1.67%] | +0.06% [-0.05%, +0.17%] | 2.49 [2.23, 2.74] |

**Table 10: Metric comparison among GSP mechanisms under different reserve and non-uniform bid-scaling settings.**

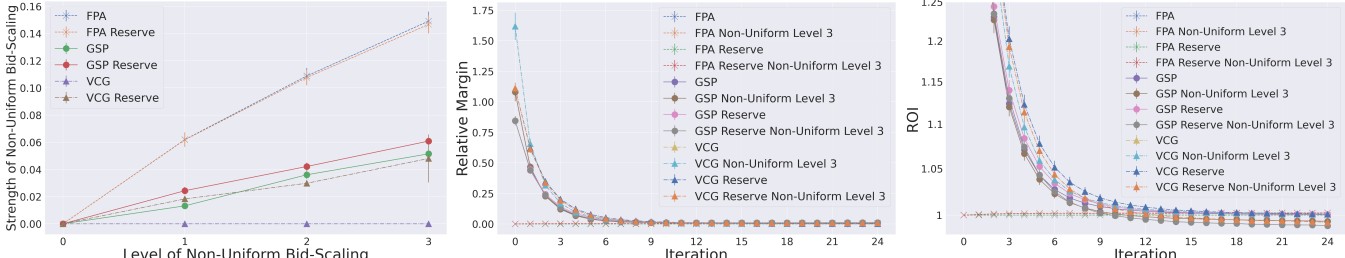

**Figure 5: The strength of non-uniform bid-scaling by level, as well as relative margin and ROI convergence for auction formats with reserves.**

