# OpenReview forum: "Non-uniform Bid-scaling and Equilibria for Different Auctions: An Empirical Study"
_ACM.org/TheWebConf/2024/Conference — TheWebConf24_

### Official Review · Reviewer_vp5E · 2023-11-08

**Novelty:** 5
**Technical Quality:** 6

**Review:**

This paper addresses the complex issue of non-uniform bid-scaling and its impact on auction equilibria across various auction formats, a topic that is highly pertinent in the realm of online advertising. The authors delve into a critical question: How does non-uniform bid-scaling influence the efficiency and outcomes of different auction mechanisms, such as first-price, Vickrey-Clarke- Groves, and generalized second-price auctions? Recognizing the limitations of current autobidding systems to adapt to such scaling strategies, the authors introduce an empirical study that simulates auction scenarios to observe the effects of bid-scaling. Through a combination of feature vector generation for auction simulation and ad candidate retrieval processes, the study evaluates the performance of these auction formats under non-uniform bid-scaling conditions. The manuscript presents a comprehensive analysis, comparing the equilibria achieved in each auction type and discussing the implications for auction theory and practice. The findings suggest that the application of non-uniform bid-scaling can significantly alter auction outcomes, challenging the traditional understanding of auction equilibria and offering new perspectives for the design of more efficient and equitable online auction systems.
Advantages:
•	The study provides a comprehensive empirical analysis of different auction formats, including VCG, GSP, and FPA, under various bid-scaling scenarios. This empirical approach is valuable because it offers real-world insights that can validate or challenge theoretical models.
•	The paper includes an analysis of the convergence properties of auto-bidding algorithms, ensuring that the empirical results are robust and reflective of what would happen in equilibrium. This is crucial for validating the practical applicability of the findings.
•	The paper presents a detailed comparison of auction formats with and without reserves and under different bid-scaling conditions. This comparative analysis is beneficial for understanding the nuances of each auction type and their respective efficiencies.
Disadvantages:
•	The empirical results section extends previous work to different auction formats, non-uniform bid-scaling, and with costs. However, it is not clear how robust these extensions are and whether they accurately capture the complexities of real-world auction scenarios.
•	The paper might be dense and complex, potentially requiring more intuitive diagrams or visual aids to enhance understanding. The process of dataset generation and simulation can be illustrated with some diagrams
Overall, despite minor flaws in the content organization, this is an excellent paper that deserves acceptance.

**Questions:**

(1) Could you provide more details on the computational complexity of the simulations and whether the current methodology scales to larger, more complex auction environments?
(2) In simulating the candidate retrieval process, a threshold for correlation scores is mentioned. How was this threshold determined, and could its calibration significantly affect the results?

**Reviewer Confidence:**

3: The reviewer is confident but not certain that the evaluation is correct

**Scope:**

4: The work is relevant to the Web and to the track, and is of broad interest to the community

---

### Official Review · Reviewer_n8bM · 2023-11-19

**Novelty:** 5
**Technical Quality:** 4

**Review:**

Summary of Main Content:
The paper conducts an empirical study on different auction formats under uniform bid-scaling and non-uniform bid-scaling in an auto-bidding world. The paper has obtained some intriguing empirical findings, which validate and broaden the theoretical results. Furthermore, the auto-bidding simulation experimental framework proposed in this paper, encompassing the generation of synthetic data, simulation of various auction formats, and emulation of different bidding strategies, lays the groundwork and provides a paradigm for subsequent related research.

Strengths:
The core issue of the paper is articulated clearly and explicitly, with a comprehensive and detailed introduction to related work. The framework for simulating auto-bidding proposed in this paper is highly innovative and engaging, laying a solid foundation and providing a paradigm for subsequent related research, and offering the potential for a closer integration of theory and application.

Weaknesses:
In terms of the drawbacks of this paper, I observe the following points:
1.The background introduction to auto-bidding is overly detailed, while there is relatively less emphasis on the introduction of the research significance pertaining to the main theme of the paper.
2.The experimental conclusions are somewhat one-dimensional; part of the findings validate theoretical results, but another part lacks theoretical evidence.
3.I believe that there is no necessity for Algorithms 1 and 2; they simply manifest the allocation and payment rules of several auction formats in pseudocode, leading to redundant content.
4.The theme of the paper encompasses both Uniform and Non-Uniform studies, yet only Non-Uniform is reflected in the title.
5.There is a formatting issue within the article, such as in line 602, where there is only one character, making this part of the layout somewhat unattractive.

Advices:
My first suggestion is to trim certain parts of the content to ensure the article is concise and focused, highlighting the main points. For instance, the development history of auto-bidding auctions is relatively less related to the main theme of the paper and should be introduced in a more succinct manner.

Results solely from the experimental aspect may not be sufficient to substantiate an entire article. It would be advisable to enrich the conclusions of the paper, ideally combining theory with experiments, such as providing theoretical explanations and validations for the experimental results.

**Questions:**

1.Some experimental conclusions in the paper lack theoretical backing. Could you provide additional evidence to support these experimental findings, particularly those that do not currently align with existing theories?
2.How do you ensure that the simulation results obtained from the experimental framework, using the data generated, are consistent with real-world scenarios?
3.Algorithms 1 and 2 simply manifest the allocation and payment rules of several auction formats in pseudocode. What is the significance of this section in the paper? Is it redundant?
4.The experiment includes both uniform bid-scaling and non-uniform bid-scaling, yet why does the title only emphasize non-uniform bid-scaling?

**Reviewer Confidence:**

3: The reviewer is confident but not certain that the evaluation is correct

**Scope:**

3: The work is somewhat relevant to the Web and to the track, and is of narrow interest to a sub-community

---

### Official Review · Reviewer_MqbD · 2023-11-21

**Novelty:** 6
**Technical Quality:** 5

**Review:**

The authors consider a typical auction model in the auto-bidding context where participating bidders are value maximizers but have some constraint over their spend.  In particular, the authors examine the case of value-maximizing bidders with return on spend constraints (i.e., the value obtained by any bidder needs to be more than her total payment, the analog of an IR constraint for the more standard model in economics of quasilinear utility bidders) when the auctioneer aims to maximize the social welfare.  Previous work has examined auction formats like the Vickrey-Clarke-Groves (VCG) auction, generalized second-price (GSP) auction, and the first-price auction (FPA) from the perspective of worst-case analysis (namely analyses of the price of anarchy of these auctions).  In contrast, this paper takes an empirical perspective, proposing a scheme to generate auction instances and measure the collected welfare and revenue of the three auctions.

The central technical contribution of the paper is a generative model for synthetic auction data which can be used to test the “average case” performance of an auction (where the average is over the randomness in the data generation).  Using this data generation framework, the authors demonstrate that, in the presence of both uniform and non-uniform bid-scaling, the FPA obtains better welfare and profit than the GSP, which is in turn better than VCG on both of these dimensions.  However, the performance of the FPA and GSP degrades with different amounts of non-uniform bid-scaling whereas VCG retains very similar levels of performance across levels of non-uniformity.

On the whole, while I am not certain that I fully expect the authors’ proposed generative model for data will see a great deal of independent use/interest, I think the results of this paper are non-trivial, interesting, and likely to be appreciated by those interested in the auto-bidding setting (which represents a large portion of the WebConf community interested in internet economics).  I think the takeaway message, namely, that the FPA performs better than VCG “on average” (at least on the data that is generated by the authors’ framework) is nice and neatly complements the theoretical analyses of these auctions which have been gaining attention in the community.  On the other hand, I think the presentation of this paper could use some improvements and there are areas for further improvement of the results themselves which would enhance future versions of this paper.  First, while return-on-spend constraints are certainly important and widely studied, another common constraint is that of budgets, which is overlooked by the authors’ analyses.  It would be interesting to see how the results change in the presence of these more elaborate constraints.  Secondly, it would be interesting to see how the results change if the model of the cost of showing ads was not “additive” over the ads (see lines 185-189).  Finally, in terms of presentation, there are some smaller issues which I outline below.

In Section 4.4, the discussion in the opening paragraph for convergence seems muddled with respect to earlier discussions in the paragraph beginning at line 86.  Namely, you justify the study of the models on synthetic data “in the average case” because “finding… equilibrium could be computationally intractable” (lines 93-94) and yet you are concerned with convergence to equilibrium in Section 4.4 in order that the results model what would happen in a practical equilibrium (which feels contrary to your discussion earlier).  I actually think this section is important and interesting, but I would suggest rewording it.

Line 304: “target overtime” -> “target over time”

Line 310: “in the decreasing” -> “in decreasing”

Line 314: “Only the ad gets clicked will be charged” -> “Only the ad that gets clicked will be charged”

Line 338: “and hence no corresponding” -> “and hence has no corresponding”

Example 3.1 could use significant rewording.

Line 622: “will be much large” -> “will be much larger”

Line 631: “auction simulation done is” -> “auction simulations done is”

Line 708: “in such setting” -> “in such a setting”

Lines 745-6: “VCG does not be simply generalized” -> “VCG does not simply generalize”

Lines 755-758: I’m not sure I understand how showing that switching to non-uniform bid-scaling decreases welfare is consistent that the FPA gives optimal welfare and profit on uniform bid-scaling.  I would suggest rewording this discussion.

[After rebuttal]  I appreciate the authors' responses to the questions/comments posed by the reviewers.

**Questions:**

Can you comment on whether or not your data generation and test setup extends easily to other possible cost models for showing ads and/or additional constraints over bidder spend?  If so, have you ran experiments with other constraints/cost models and are the results similar to those in your paper (e.g., in terms of relative order of performance)?

**Reviewer Confidence:**

3: The reviewer is confident but not certain that the evaluation is correct

**Scope:**

4: The work is relevant to the Web and to the track, and is of broad interest to the community

---

### Official Review · Reviewer_GQAC · 2023-11-23

**Novelty:** 6
**Technical Quality:** 6

**Review:**

The paper provides an empirical study of different auction formats and bid-scaling strategies for a setting with value-maximizing auto bidders. The study used synthetic datasets.

Advantages:
1) The paper provides empirical findings that complement existing worst-case theoretical results. The average-case empirical findings (in a setting without independence and symmetry) agree with existing worst-case theoretical results using the price of anarchy notion.

2) The paper provides additional empirical findings to show that Generalized Second-Price auction has better welfare and profit than Vickrey-Clarke-Groves auction. These are not implied by existing worst-case theoretical results. Overall, the findings show the complete ordering of First-Price auction, then Generalized Second-Price auction, then Vickrey-Clarke-Groves auction in terms of both welfare and profit and for both uniform and non-uniform bid-scaling bidding strategies.

3) The paper’s simulation framework may be applicable in other settings. In particular, the authors suggest a multi-channel setting where each channel can adopt its own auto-bidding strategy.

**Questions:**

What are the intervals below the average numbers reported in Tables 2-10?

**Reviewer Confidence:**

3: The reviewer is confident but not certain that the evaluation is correct

**Scope:**

4: The work is relevant to the Web and to the track, and is of broad interest to the community

---

### Decision · Program_Chairs · 2024-01-22

**Decision:**

Accept

**Comment:**

The paper presents an empirical study of autobidding strategies where individual auctions can be First Price, GSP and VCG. To do this, they propose a method for synthetic data generation and under a variety of settings compare the welfare and revenue under the different auction format, both for autobidding strategies where values are uniformly scaled down, or where they can scale down non-uniformly.

 The reviewers found the empirical evaluation to be comprehensive, appreciated the value of the synthetic data generating process. Overall the recommendation is to accept the paper. There were several suggestions to improve the presentation; we encourage the authors to include these in the camera ready.